# Production of Phenyllactic Acid from *Porphyra* Residues by Lactic Acid Bacterial Fermentation

**Chung-Hsiung Huang \***, **Wei-Chen Chen, Yu-Huei Gao, Hsin-I Hsiao and Chorng-Liang Pan \***

Department of Food Science, National Taiwan Ocean University, Keelung 20224, Taiwan; s1020094@gm.pu.edu.tw (W.-C.C.); ab123k6@gmail.com (Y.-H.G.); hi.hsiao@ntou.edu.tw (H.-I.H.)

\* Correspondence: huangch@mail.ntou.edu.tw (C.-H.H.); b0037@mail.ntou.edu.tw (C.-L.P.);
  Tel.: +886-2-2462-2192 (ext. 5115) (C.-H.H.); +886-2-2462-2192 (ext. 5116) (C.-L.P.)

**Abstract:** The concept of algae biorefinery is attracting attention because of the abundant valuable compounds within algal biomass. Phenyllactic acid (PhLA), a broad-spectrum antimicrobial organic acid that can be produced by lactic acid bacteria (LAB), is considered a potential biopreservative. In this study, a cascading biorefinery approach was developed to harvest PhLA from *Porphyra* residues by LAB fermentation. LAB strains were cultivated in de Man, Rogosa and Sharpe (MRS) broth to screen their ability to produce PhLA. As the strains of *Lactobacillus plantarum* KP3 and *L. plantarum* KP4 produced higher concentrations of PhLA at 0.09 mg/mL, these two strains were employed for fermentation. After phycobiliprotein extraction, the *Porphyra* residues, ultrafiltration eluate, phenylalanine (Phe) and yeast extract with a volume of 20 mL were inoculated with LAB strain KP3 and fermented at 37 °C for 120 h. The PhLA content of the fermented broth was 1.86 mg. To optimize the biorefinery process, the ultrafiltration eluate was replaced by commercial cellulase. Up to 4.58 mg of PhLA, which was 2.5 times greater than that produced from KP3 cultured in MRS broth, could be harvested. Taken together, the findings provide an optimized process for LAB fermentation, which is shown to be a feasible algae biorefinery approach to obtaining PhLA from *Porphyra* residues.

**Keywords:** algae; cellulase; fermentation; lactic acid bacteria; phenyllactic acid; *Porphyra* residues



## 1. Introduction

Algae biomass consists of many components, including carbohydrates, lipids, and proteins, which makes it a feasible feedstock for conversion into various types of bioproducts [1]. Algae biorefinery involves processing worthless algae biomass or residues for the production of valuable products and reducing organic wastes [1]. Effectively using algae residues not only reduces waste for achieving environmental sustainability, but also decreases the cost of production and increases the added value of by-products or final products [1]. Many studies employed macroalgae as biological substrates to extract pigments, proteins, saccharides, and other derivative products, but algae residues are yielded after the process of extraction. The composition of algae residues, which is dependent on the alga strain and extraction method, has a considerable impact on the bioreaction and downstream process of biorefinery [2]. The valuable products from algae biorefinery can be produced via different biorefinery processing methods [3]. In the fermentation approach, microorganisms use carbohydrates and proteins within pretreated algal biomass such as carbon and nitrogen sources for the growth and production of metabolites [4]. Pretreatment of algae biomass refers to the disruption of the algae cell wall, which is primarily composed of glycoproteins and polysaccharides, prior to extraction [5]. In our recent study, the crude enzyme solution produced by marine bacterial strains successfully disrupted the *Porphyra* cell wall for phycobiliprotein extraction [6].

Phenyllactic acid (PhLA) is one of the organic acids that has been of great interest in recent years due to the increased demands for food, cosmetic, pharmaceutical, and chemical industries [7]. PhLA has broad-spectrum antimicrobial activity against a wide

species of Gram-positive and Gram-negative bacteria as well as some fungal species [8]. Therefore, it could potentially be employed as a food preservative or feed additive. PhLA can be generated naturally as metabolites by various microorganisms [8]. In general, PhLA is a by-product of phenylalanine (Phe) metabolism in lactic acid bacteria (LAB), in which Phe is transaminated to phenylpyruvic acid by aromatic aminotransferase and further reduced to PhLA by dehydrogenase and nicotinamide adenine dinucleotide [7]. Studies have shown that many strains of LAB can produce PhLA in de Man, Rogosa and Sharpe (MRS) medium, but the concentration of PhLA is not high, only about 94 mg/L [9]. Although microorganisms that produce PhLA with high efficacy could be created by genetic engineering technology, natural LAB strains are more attractive in the food industry due to the cost and safety benefits [10]. In LAB fermentation, several critical factors, including the concentrations of the microbial cells, PhLA precursors, carbon and nitrogen sources, the environmental pH, and reaction time, significantly influence the bioconversion process and the subsequent yield of PhLA and conversion ratio [7]. Obstacles to using this approach still remain, such as the lack of microbial strains with high efficiency and the lack of effective design methods for the optimization of these processes [7].

*Porphyra*, one of the most cultured red algae, is gaining in economic importance, and is cultivated, harvested, dried, processed, and consumed in large quantities in East Asia [11]. Compared with other algae, *Porphyra* is rich in carbohydrates and protein, with a carbohydrate content of 40 to 76% and a protein content of 7 to 50% [11]. In our recent study, the crude enzyme solutions were produced by two marine bacterial strains. By employing the crude enzyme solutions, an enzyme-assisted method for phycoerythrin (PE) and phycocyanin (PC) extraction from *Porphyra* was developed [6]. To date, no literature pertaining to the production of PhLA from algae is available. Therefore, we aimed to develop a cascading biorefinery process to harvest PhLA from *Porphyra* residues by LAB fermentation. First, the feasibility of PhLA production from *Porphyra* residues was evaluated, and the LAB strains were screened to determine their capacity for PhLA production. Second, the composition of *Porphyra*-residue-containing formula was modified for the successful fermentation and production of PhLA. Finally, pretreatment of *Porphyra* residues with cellulase was introduced to optimize the fermentation process and to increase the yield of PhLA.

## 2. Materials and Methods

### 2.1. Chemicals and Reagents

All the chemicals and reagents were purchased from Sigma-Aldrich Chemical Co. (St. Louis, MO, USA) and Panreac Química SLU (Castellar del Vallès, Barcelona, Spain) unless otherwise stated. The reagents for bacterial culture were purchased from Difco Laboratories (Detroit, MI, USA). *Porphyra* sp., purchased from Xin meng cheng company (Penghu, Taiwan), was ground, sieved (0.38 mm pore size), and stored at 4 °C before use. Cellulase AP3 was purchased from Amano Enzyme Inc. (Nagoya, Aichi, Japan).

### 2.2. Preparation of Porphyra Residues and Untrafiltration Elute

The process of phycobiliprotein extraction was accomplished according to our previous study [6]. Briefly, *Porphyra* powder was added to a crude enzyme solution. The mixture was homogenized and ultrasonicated followed by incubation at 26 °C for 72 h. After centrifugation at 12,000× *g* for 20 min, the supernatants were subjected to ultrafiltration (UF) with a 100 kDa hollow fiber membrane (UFP-100-E-6A, GE Healthcare, Chicago, IL, USA) as UF eluate. The eluate with molecular weight higher than 100 kDa was named as UF > 100 kDa, and that lower than 100 kDa was named as UF < 100 kDa. The precipitate was collected as *Porphyra* residues for further experiments.

### 2.3. Screening the Strains of LAB for PhLA Production

Four LAB strains isolated from commercial fermented foods, including *Lactobacillus plantarum* KP3, *L. plantarum* KP4, *Leuconostoc mesenteroides* K8 and *L. paracasei* subsp. *para-*

*casei* DP2, were cultivated in MRS broth at 37 °C for 72 h. The cultivated medium and cell-free supernatants were collected every six hours to determine the LAB count, PhLA concentration, optical density (O.D.) at 600 nm, and pH value.

### 2.4. Fermentation of Porphyra Residues by LAB

The experiments were accomplished according to a recent report [12] with some modifications to comprehensively investigate the impact of reaction parameters on LAB fermentation and PhLA production. In this study, *Porphyra* residues were pretreated with 0, 50, 100, 500, 1000, 2000, or 4000 U cellulase AP3 at 37 °C for 0 to 48 h for enzymatic saccharification. For fermentation, pretreated or untreated *Porphyra* residues (2%, 4%, or 8%) were added with or without UF < 100 kDa (8.21%), Phe (1.36% or 6.8%), and/or yeast extract (0.5% or 2.5%), and inoculated with cultivated KP3 or KP4 (1%; v/v) for 228 h. The cultivated medium and cell-free supernatants were collected every six hours to determine the LAB count and pH value. After centrifugation at 12,000× *g* for 20 min, the supernatants were lyophilized for extraction and determination of PhLA, lactic acid, and Phe. To clearly show the results in the limited space, not all of the data collected every six hours during the period of fermentation are shown in the following figures.

### 2.5. Determination of Total Sugar and Reducing Sugar Contents

Total soluble sugar was determined based on the method described by Dubois [13]. The supernatants were mixed with phenol and $H_2SO_4$ and incubated at room temperature for four hours. The absorbance was read at 490 nm wavelength on a UV spectrophotometer. Galactose was used as standard, and the total sugar contents within tested samples were calculated relative to standard. According to the method reported by Miller et al. [14], the supernatant was collected and mixed with an equal volume of 3,5-dinitrosalicylic acid (DNS) reagent (Sigma-Aldrich, St. Louis, MO, USA). The mixture was heated at 100 °C for 10 min, and cooled prior to being mixed with an equal volume of deionized water. The mixture was measured at an absorbance at 546 nm, and the reducing sugar contents were calculated. The standard curve was accomplished using simple sugar galactose.

### 2.6. Extraction and Determination of PhLA, Lactic Acid, and Phe

According to the methods reported from previous studies, the lyophilized powder was dissolved in formic acid (pH 2) and mixed with an equal volume of ethyl acetate for 15 min [8,15]. The mixture was dried using $Na_2SO_4$ and concentrated in a rotary evaporator. The dried residues were reconstituted with $H_2SO_4$ and analyzed in an HPLC system fitted with an ICSep Coregel Ion-300 column (Transgenomic) using $H_2SO_4$ as the mobile phase at 0.4 mL/min and a refractive index (RI) detector. Commercial PhLA and lactic acid (Sigma Chemical Co., St Louis, MO, USA) were used as references. According to the methods reported by Li et al., Phe was determined in an HPLC system fitted with a Bio-bond C18 column (Dikma), using the mobile phase consisting of a gradient mixture of mobile phases A (methanol + 0.05% trifluoroacetic acid) and phase B (double-distilled water + 0.05% trifluoroacetic acid) at 0.5 mL/min and a UV-visible detector set to 210 nm [16]. Commercial Phe (Sigma Chemical Co., St Louis, MO, USA) was used as reference (retention time = 18 min).

### 2.7. Statistical Analysis

Data were statistically analyzed using IBM® SPSS Statistic® (IBM, Armonk, NY, USA, 2015). One-way analysis of variance (ANOVA) was used to determine statistical differences between sample means, with the level of significance set at $p < 0.05$. Multiple comparisons of means were achieved using Duncan's test. All data are expressed as mean ± standard deviation.

## 3. Results and Discussion

### 3.1. The Feasibility of PhLA Production from Porphyra Residues by Fermentation Process

In our recent study, *Porphyra* were incubated within crude enzyme solution with homogenization and ultrasonication followed by the ultrafiltration (UF) process. The valuable phycoerythrin (PE) and phycocyanin (PC) were extracted and mainly existed in UF eluate with a molecular weight higher than 100 kDa (UF > 100 kDa) [6]. To determine the feasibility of employing *Porphyra* residues as biomass for PhLA production, the total sugar and reducing sugar contents within *Porphyra* enzymatic hydrolysates, *Porphyra* UF eluate, and unfermented *Porphyra* residues were investigated. Higher concentrations of total sugar and reducing sugar were detected in *Porphyra* enzymatic hydrolysates and *Porphyra* UF eluate than in unfermented *Porphyra* residues (Table 1). In parallel, the LAB strains employed for *Porphyra* residue fermentation were screened by determining their capacity for PhLA production. The pH values of cultured MRS broth of these four LAB strains dropped rapidly from 6.5 to 3.75 during the logarithmic growth phase at 0 to 18 h (Figure 1), suggesting that a large amount of organic acids was produced by these LAB strains. After culturing for 18 h, these LAB strains entered a stationary phase, and the number of these strains increased to $10^8$ colony forming units (CFU)/mL (Figure 1). In line with the results of previous studies, these LAB strains began to produce PhLA during the stationary phase to inhibit the growth of other bacteria [7,17]. Compared with the other strains, KP3 and KP4 produced higher levels of PhLA at the concentration of 0.09 mg/mL, which agrees with the results of previous studies [9,18]. The numbers of KP3 and KP4 were much higher than $10^8$ CFU/mL. Therefore, KP3 and KP4 were selected as candidate strains for *Porphyra* residue fermentation.

**Table 1.** Analysis of total sugar, reducing sugar, lactic acid, and phenyllactic acid (PhLA) concentration of Porphyra enzymatic hydrolysates, Porphyra ultrafiltration eluate, and unfermented and fermented Porphyra residues. Each value is mean ± standard deviation (n = 3).

| Sample | Total Sugar (mg/mL) | Reducing Sugar (mg/mL) | Lactic Acid (%) | PhLA(%) |
|---|---|---|---|---|
| *Porphyra* enzymatic hydrolysates [1] | 14.22 ± 1.27 | 1.12 ± 0.11 | - | - |
| Ultrafiltration (UF) > 100 kDa [2] | 14.02 ± 1.38 | 1.10 ± 0.10 | - | - |
| UF < 100 kDa [3] | 11.87 ± 1.20 | 0.86 ± 0.10 | - | - |
| Unfermented *Porphyra* residues [4] | 4.80 ± 0.19 | 0.42 ± 0.06 | - | - |
| KP4 fermented *Porphyra* [5] | 0.06 ± 0.00 | 0.04 ± 0.01 | 0.13% | N.D. [7] |
| KP3 fermented *Porphyra* [6] | 0.05 ± 0.00 | 0.03 ± 0.00 | 0.13% | N.D. |

[1] Porphyra enzymatic hydrolysates: 2% Porphyra powder in 100 mL crude enzyme solution for hydrolysis at 26°C for 24 h. [2] UF > 100 kDa: UF eluate of *Porphyra* enzymatic hydrolysates with molecular weight higher than 100 kDa. [3] UF < 100 kDa: UF eluate of *Porphyra* enzymatic hydrolysates with molecular weight lower than 100 kDa. [4] Unfermented Porphyra residues: Almost 4% Porphyra residue powder in distilled $H_2O$. [5] KP4 fermented Porphyra: Unfermented Porphyra residues (4%) inoculated 1% KP4 and fermented for 24 h. [6] KP3 fermented Porphyra: Unfermented Porphyra residues (4%) inoculated 1% KP3 and fermented for 24 h. [7] N.D.: Not detected.

### 3.2. PhLA Production from Porphyra Residues by Fermentation with KP3 and KP4 Strains

After fermentation of *Porphyra* residues (4%) with either KP3 or KP4 for 24 h, the number of LAB increased up to $10^8$ CFU/mL, and the total sugar and reducing sugar contents markedly depleted (Table 1). However, only a small amount of lactic acid and no PhLA were detected in the fermented broth (Table 1). The above results suggested that KP3 and KP4 strains can use *Porphyra* residues for fermentation. The limited production of lactic acid and PhLA may have resulted from a lack of nutrients or PhLA precursors, such as Phe. Therefore, the impact of *Porphyra* residue concentration on PhLA production was further explored. As shown in Figure 2, the amount of KP3 and KP4 increased and gradually decreased after fermentation of 8% *Porphyra* residues for 72 and 96 h, respectively. The number of bacteria obviously decreased after fermentation of 2% *Porphyra* residues for 12 h, revealing that 8% *Porphyra* residues provides much more nutrients and a fermentation environment with a suitable pH for LAB growth compared with 2% *Porphyra* residues. The peak phase period of LAB cultured with 8% *Porphyra* residues

was persistent from 6 to 120 h post-fermentation, which is much longer than that cultured with 2% *Porphyra* residues and that in MRS medium. Thus, 8% was determined as the highest ratio of *Porphyra* residues for LAB fermentation. Noticeably, there was no PhLA within the supernatants after fermentation for 0 to 120 h. To enable biorefinery and to reduce waste, UF < 100 kDa was used as the additive to provide a carbon source and trace elements for *Porphyra* residue fermentation. Based on the results of our previous study, we calculated that 22.58 g of lyophilized UF < 100 kDa and 5.5 g of *Porphyra* residues were yielded in 1 L of *Porphyra* enzymatic hydrolysates [6]. Therefore, the ratio of 2% *Porphyra* residues and 8.21% UF < 100 kDa was employed for the fermentation process. The ratio of Phe (1.36%) and yeast extract (0.5%) in the *Porphyra*-residue-containing formula for fermentation was determined according to the results of previous studies showing higher Phe conversion efficiency and better growth of LAB [19,20]. To optimize the formula for PhLA production, UF < 100 kDa (8.21%), Phe (1.36%), and/or yeast extract (0.5%) as the nitrogen source were added into 2% *Porphyra* residues for fermentation. Compared with that, with only UF < 100 kDa, Phe, or yeast extract added, KP3 cultured in *Porphyra* residues (2%) with UF < 100 kDa (8.21%), Phe (1.36%), and yeast extract (0.5%) produced PhLA after fermentation for 12 to 120 h in a time-dependent manner (Figure 3). The PhLA content within the fermented broth was 1.86 mg at 120 h, and the rate of conversion from Phe to PhLA was 40.46% (Figure 3D). As shown in Figure 3D, the concentration of lactic acid and the pH value did not significantly change during the fermentation period. Phe was not yet completely converted, suggesting that the ratio of added Phe and yeast extract provided sufficient PhLA precursor and nitrogen for KP3 fermentation. To further evaluate the impact of Phe and yeast extract on PhLA production, the concentrations of Phe and yeast extract were increased to 6.8% and 2.5%, respectively. However, the growth of LAB strains was not observed (data not shown), probably due to high osmotic pressure.

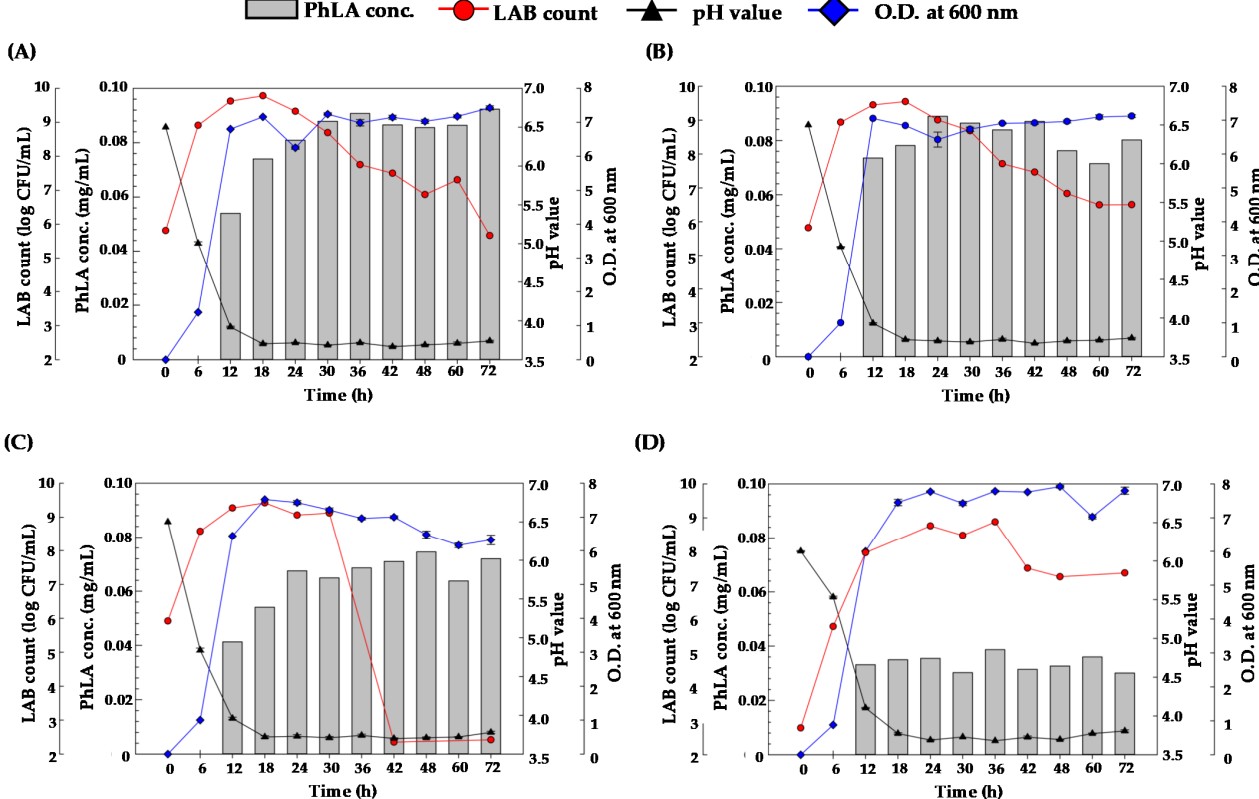

**Figure 1.** The lactic acid bacteria (LAB) count, PhLA concentration, optical density (O.D.) at 600 nm, and pH value of cell-free supernatant of (**A**) *Lactobacillus plantarum* KP4, (**B**) *L. plantarum* KP3, (**C**) *Leuconoctoc mesenteroides* K8, and (**D**) *L. paracasei* subsp. paracasei DP2 in de Man, Rogosa and Sharpe (MRS) medium incubated at 37 °C for 0–72 h.

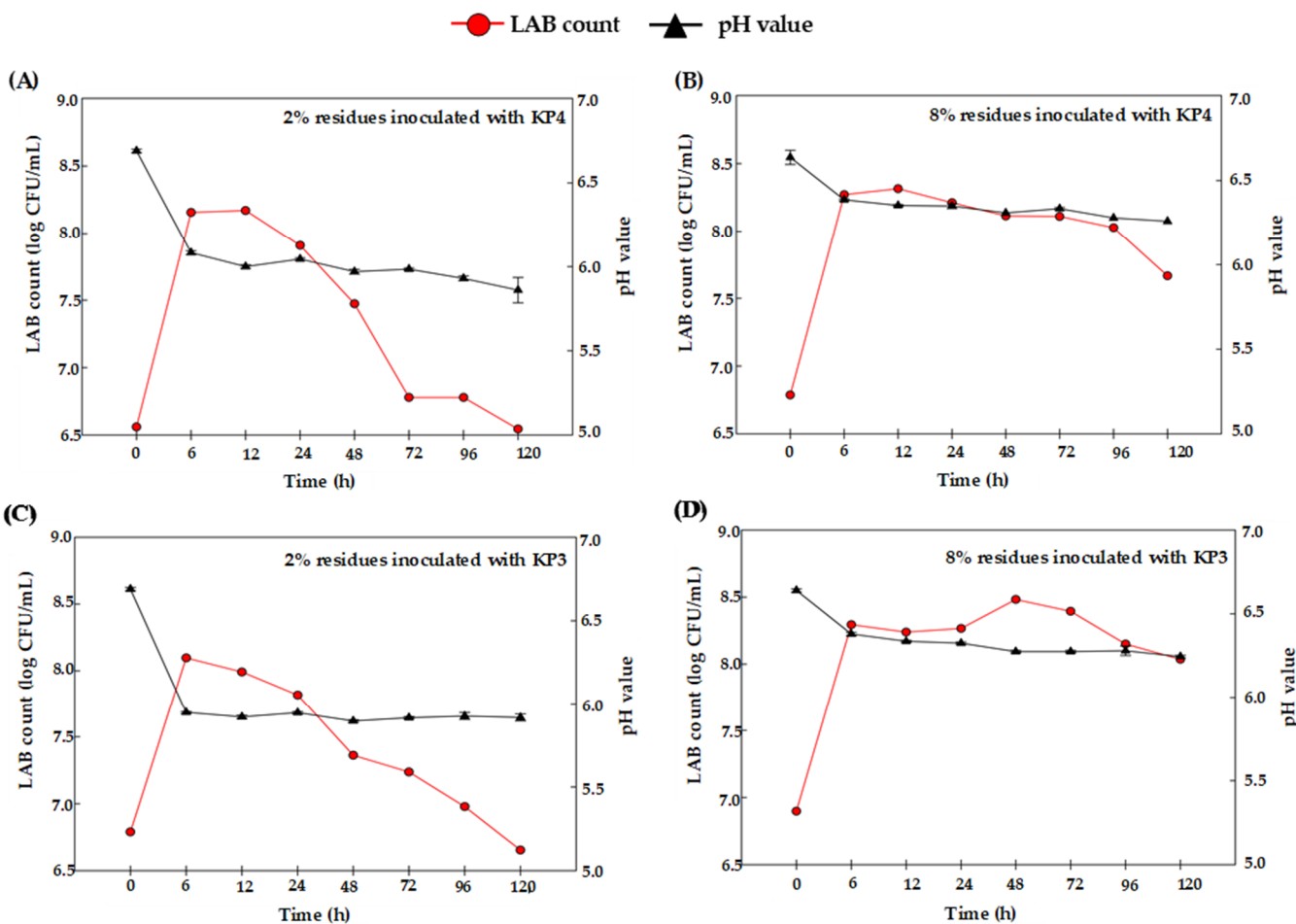

**Figure 2.** The LAB count and pH value of (**A,C**) 2% and (**B,D**) 8% *Porphyra* residues fermented with either KP4 or KP3 at 37 °C for 0 to 120 h, respectively.

### 3.3. Pretreatment of Porphyra Residues with Cellulase for PhLA Production by LAB Fermentation

To efficiently elevate the level of available carbon source, commercial cellulase was employed for the pretreatment of *Porphyra* residues. After the incubation of 2% *Porphyra* residues with cellulase, the levels of total sugar and reducing sugar considerably increased within 0 to 12 h and gradually increased over 12 to 48 h (Figure 4). Notably, *Porphyra* residues pretreated with 100, 500, 1000, 2000, and 4000 U of cellulase produced approximately 0, 10, 20, 40, and 80 mg/mL of reducing sugars, respectively. Accordingly, the higher the level of cellulase employed, the greater the amount of total sugar and reducing sugar produced (Figure 4). Therefore, 2% *Porphyra* residues was incubated with cellulase (4000 U) followed by inoculation with 1% KP3 strain for fermentation. However, the number of KP3 markedly decreased, and no viable KP3 was detected after fermentation for 72 h (Figure 5). This finding suggested that the osmotic pressure was too high due to the high concentration of total sugar for KP3 to grow and survive. To optimize the pretreatment process of *Porphyra* residues and reduce the consumption of commercial cellulase, 8% *Porphyra* residues were incubated with cellulase (50 to 2000 U) followed by KP3 inoculation and fermentation. As shown in Figure 6, approximately 8.2, 8.2, 8.5, 9.6, and 9.2 log CFU/mL of KP3 were detected after the fermentation of *Porphyra* residues pretreated with 50, 250, 500, 1000, and 2000 U of cellulase, respectively. The higher the level of cellulase employed for pretreatment, the higher the counts of LAB and the lower the pH value (Figure 6). Taken together with the results in Figures 2, 4 and 6, a positive correlation was observed between the concentration of *Porphyra* residues and the LAB counts. Notably, the pH value of commercial cellulase AP3 is 4.5. Therefore, a lower pH

in *Porphyra* residues was observed when pretreated with more cellulase AP3. To decrease the cost of PhLA production and to avoid other factors influencing LAB fermentation, other reagents or buffer solutions for pH adjustment were not employed in this study. As the pH value and growth curve of KP3 in the fermentation of 8% *Porphyra* residues pretreated with cellulase (2000 U) were similar to that cultured in MRS broth, the number of KP3, pH value, and levels of PhLA, Phe, and lactic acid were monitored within 0 to 120 h post-fermentation. Before fermentation, the total sugar and reducing sugar concentrations were 50.19 and 32.39 mg/mL, respectively. The number of KP3 increased to the highest (8.92 log CFU/mL) at 24 h and rapidly declined 48 h post-fermentation (Figure 7). After fermentation for 12 h, the concentration of PhLA was 0.087 mg/mL (Figure 7), which is similar to that of KP3 cultured in MRS broth for 24 h (Figure 1B). The concentration of PhLA and lactic acid increased in a time-dependent manner. The highest concentration of PhLA was detected at 120 h post-fermentation at 0.229 mg/mL, which is about 2.5 times greater than that produced from KP3 cultured in MRS broth (Figure 7). Although the concentration of Phe was inversely proportionate to that of PhLA, the concentration of Phe remained at 16.82 mg/mL after fermentation for 120 h. Notably, the concentration of Phe or other Phe-rich nitrogen sources may alter the production of PhLA by LAB fermentation. Further investigation will be required to elucidate this information.

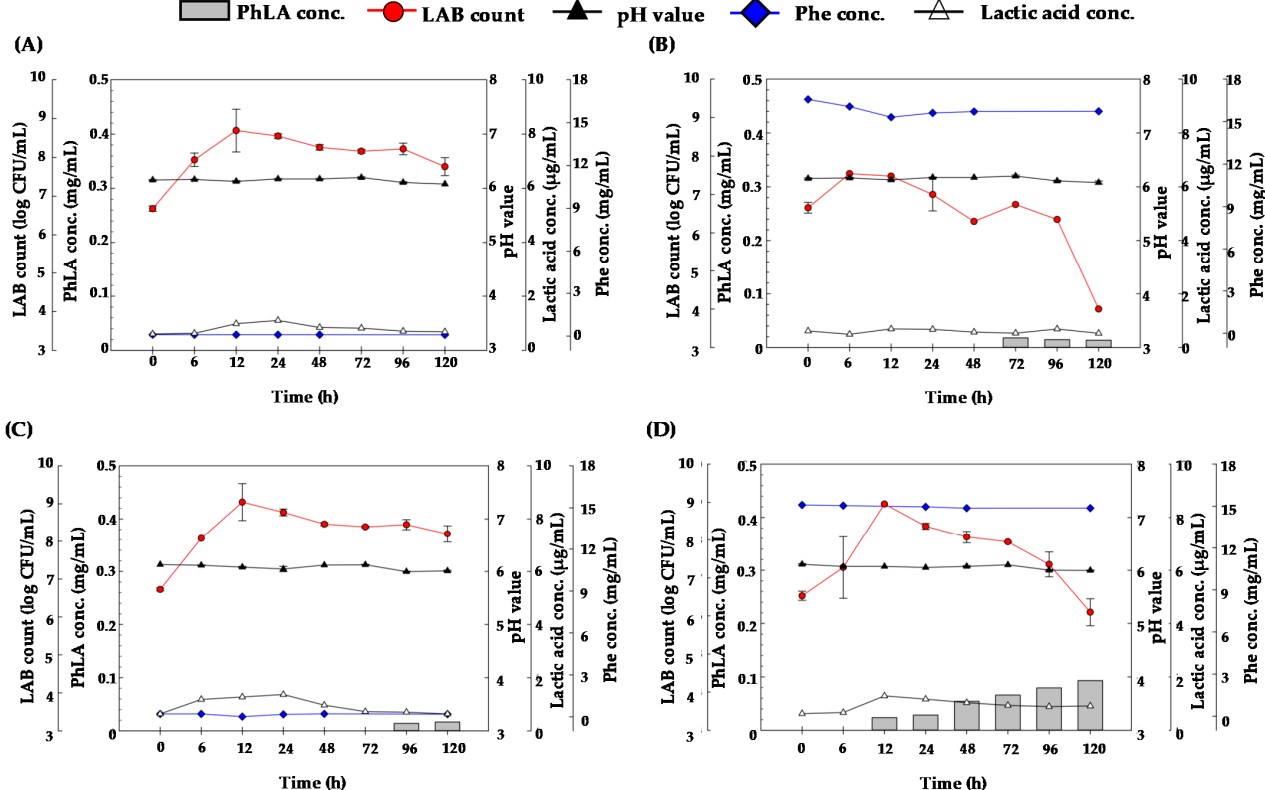

**Figure 3.** The LAB count, pH value, and concentrations of PhLA, lactic acid and phenylalanine (Phe) of (**A**) *Porphyra* residues (2%) with UF < 100 kDa (8.21%) and either (**B**) Phe (1.36%), (**C**) yeast extract (0.5%), or (**D**) both Phe (1.36%) and yeast extract (0.5%) and fermented with KP3 at 37°C for 0 to 120 h.

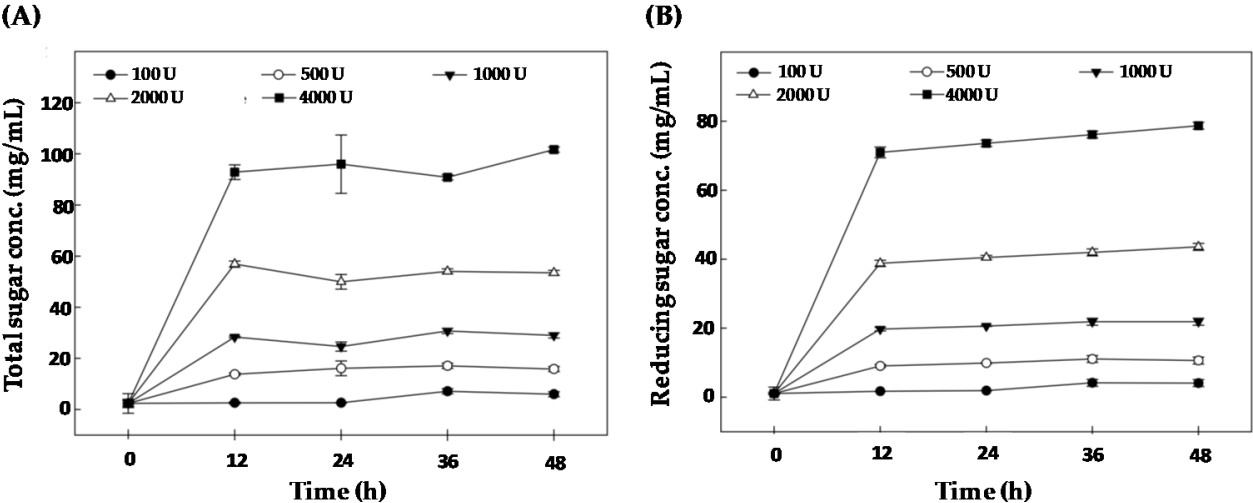

**Figure 4.** Concentrations of (**A**) total sugar and (**B**) reducing sugar within 2% *Porphyra* residues hydrolyzed by different cellulase units of commercial cellulase AP3 for 0 to 48 h.

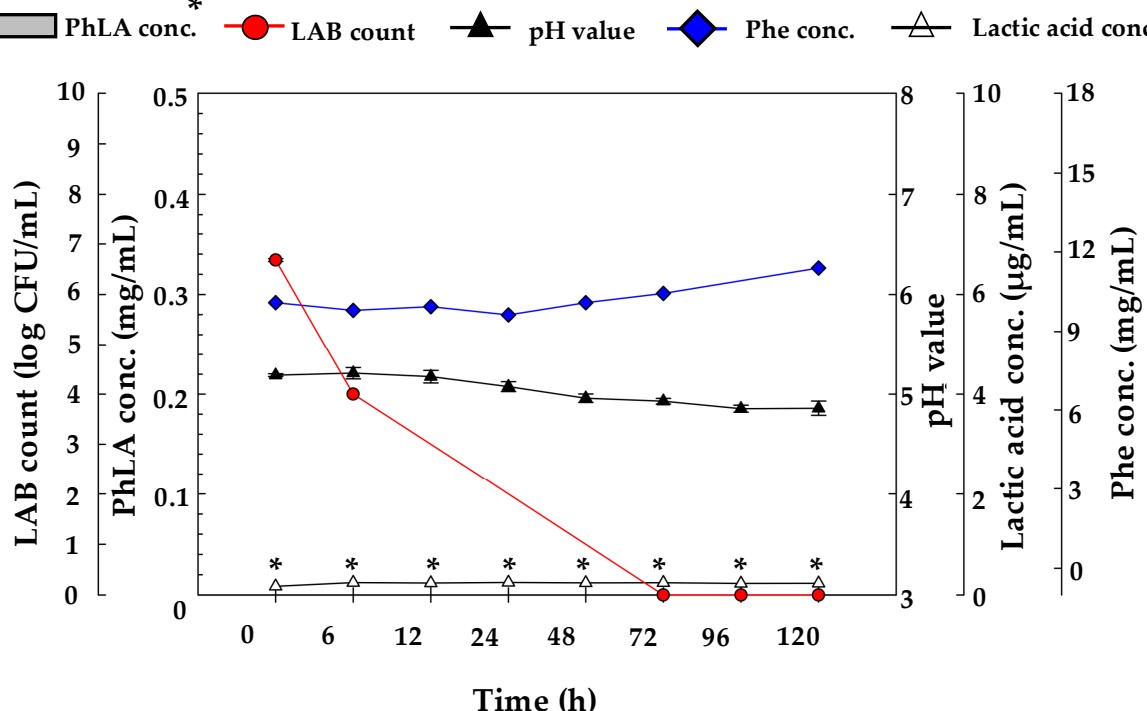

**Figure 5.** After pretreatment of *Porphyra* residues with commercial cellulase (4000 U), the LAB count, pH value, and concentration of PhLA, lactic acid, and Phe of *Porphyra* residues (2%) with the addition of UF < 100 kDa (8.21%), Phe (1.36%), and yeast extract (0.5%) and fermented with KP3 at 37 °C for 0 to 120 h. *: PhLA was not detected at the anticipated time points.

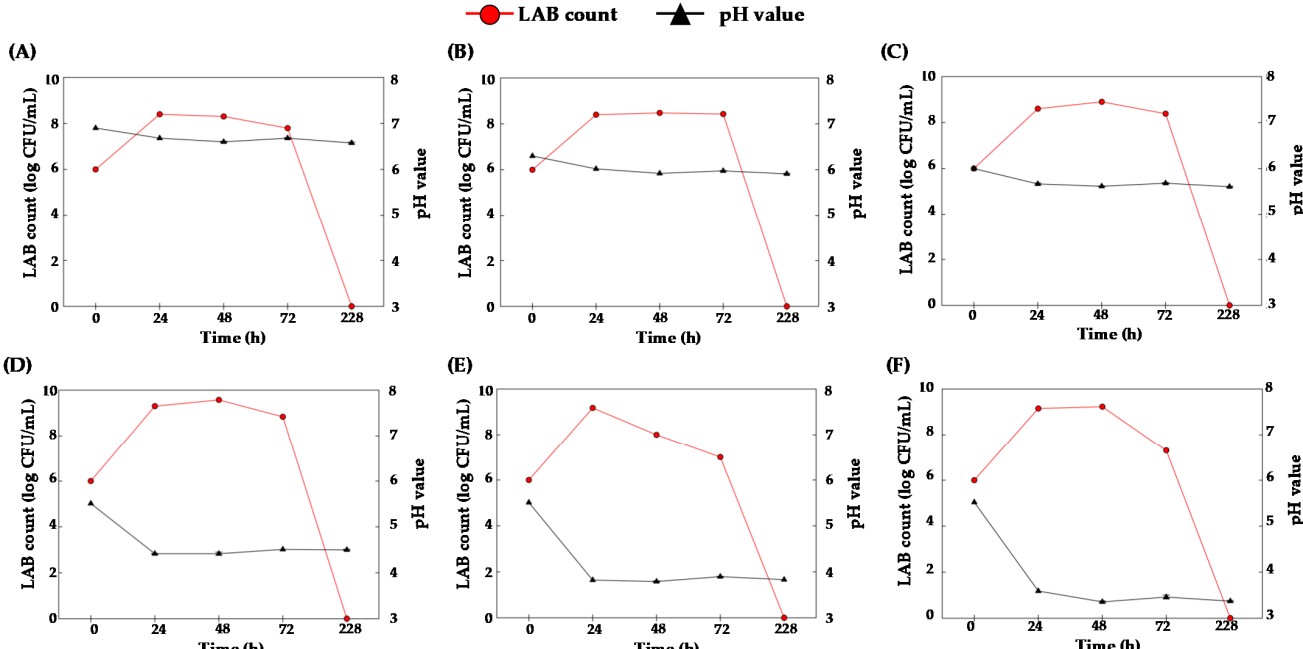

**Figure 6.** After pretreatment of *Porphyra* residues (8%) with commercial cellulase at (**A**) 50 U, (**B**) 250 U, (**C**) 500 U, (**D**) 1000 U, and (**E**) 2000 U, the LAB count and pH value of 8% *Porphyra* residues fermented with KP3 at 37 °C for 0 to 228 h are shown. (**F**) The LAB count and pH value of 8% *Porphyra* residues pretreated with 2000 U commercial cellulase and fermented with KP4 at 37 °C for 0 to 228 h.

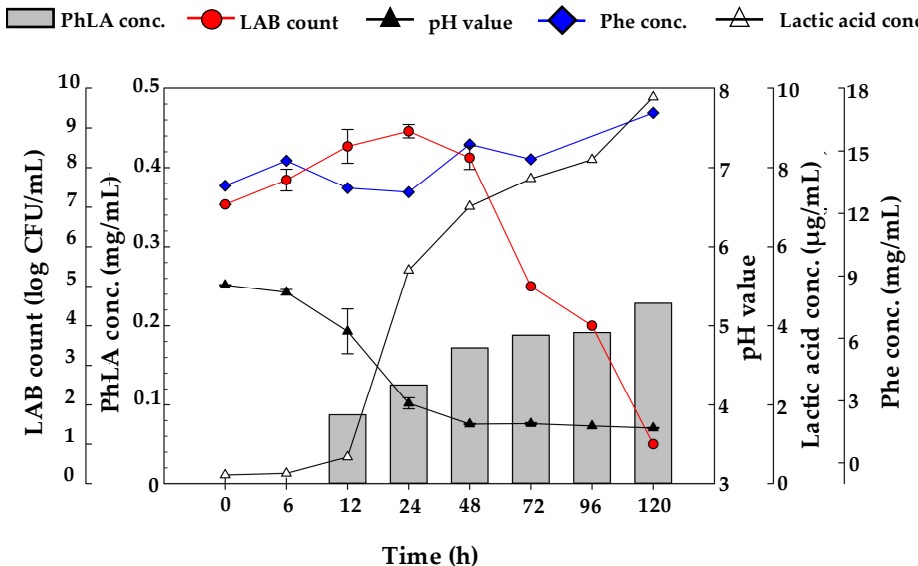

**Figure 7.** After pretreatment of *Porphyra* residues by commercial cellulase (2000 U), the LAB count, pH value, and concentration of PhLA, lactic acid and Phe in 8% *Porphyra* residues with the addition of Phe (1.36%) and yeast extract (0.5%) and fermented with KP3 at 37 °C for 0 to 120 h.

## 4. Conclusions

So far, no study pertaining to the production of PhLA from algae has been published. With respect to the use of biomass, only one study used dilute acid-pretreated sorghum bagasse as a lignocellulosic feedstock for PhLA production using a recombinant *Escherichia coli* strain expressing phenylpyruvate reductase. However, the PhLA yield was reduced by 35% compared with the filter paper hydrolysate during fermentation with the enzymatic hydrolysate of sorghum bagasse as a carbon source [21]. Recently, we developed an enzyme-

assisted method for cell-wall disruption and PE and PC extraction from *Porphyra*. However, residual *Porphyra* biomass and UF eluate (UF < 100 kDa) were yielded. In this study, we applied *Porphyra* residues and UF < 100 kDa to harvest PhLA, a valuable biopreservative, via LAB fermentation. By screening the strains of LAB and modifying the formula of the carbon and nitrogen source, PhLA precursor, cellulase, and fermentation time, the process for PhLA production was optimized. Up to 2.5 times higher the amount of PhLA was harvested than that produced from KP3 cultured in MRS broth. Taken together, integration of upstream phycobiliproteins extraction and downstream PhLA production from *Porphyra* provides a novel biorefinery process to enhance eco-friendly *Porphyra*-based bioeconomy.

**Author Contributions:** Conceptualization, methodology, investigation, writing—original draft preparation, and review and editing: W.-C.C. and Y.-H.G.; formal analysis, investigation, and editing: H.-I.H.; review and editing, supervision, project administration, and funding acquisition: C.-H.H. and C.-L.P. All authors have read and agreed to the published version of the manuscript.

**Funding:** This research was supported by the grants from the Ministry of Science and Technology (MOST 108-2221-E-019-039-MY2 and MOST 109-2320-B-019-007-MY3).

**Institutional Review Board Statement:** Not applicable.

**Informed Consent Statement:** Not applicable.

**Data Availability Statement:** Data sharing not applicable.

**Conflicts of Interest:** The authors declare no conflict of interest.

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
