# Peer review of "Production of Phenyllactic Acid from Porphyra Residues by Lactic Acid Bacterial Fermentation"

_processes, doi:10.3390/pr9040678_

Round 1
Reviewer 1 Report
This manuscript presents PhLA production using LAB strains. The production process was optimized on Porphyra Residues.
In Figure 6, after pretreatment of Porphyra residues (8%) with commercial cellulase at (A) 50 U, (B) 250 U, (C) 500 U, (D)1,000 U, (E) 2,000 U.The data showed that with the increase of cellulase for 50U, 250U and 500U, the pH value in 0h are decreased, but at 1000U and 2000U the pH value is nearly the same, what causes the pH change and why not adjust the initial pH to a suitable pH for the subsequent LAB fermentation?
The relationship between the concentration of sugar Porphyra Residues and the LAB count should be shown more clearly.
Author Response
Response to the Reviewers' comments:
We have studied carefully the comments from the Reviewers and revised the manuscript by taking all issues mentioned in the Reviewers into consideration. We highlighted all the revisions in red in the revised manuscript. The revised manuscript has undergone English language editing by MDPI. The text has been checked for correct use of grammar and common technical terms, and edited to a level suitable for reporting research in a scholarly journal. Following is a detailed point-by-point explanation on how we have addressed all the Reviewers’ concerns. The Reviewer’s comments are in black, and our responses are in blue.
Reviewer 1: This manuscript presents PhLA production using LAB strains. The production process was optimized on Porphyra Residues.
- In Figure 6, after pretreatment of Porphyra residues (8%) with commercial cellulase at (A) 50 U, (B) 250 U, (C) 500 U, (D)1,000 U, (E) 2,000 U.The data showed that with the increase of cellulase for 50U, 250U and 500U, the pH value in 0h are decreased, but at 1000U and 2000U the pH value is nearly the same, what causes the pH change and why not adjust the initial pH to a suitable pH for the subsequent LAB fermentation?
The authors thank the Reviewer for pointing out the important issue. Since the pH value of commercial cellulase AP3 is 4.5. Therefore, a lower pH in Porphyra residues was observed when pretreated with more cellulase AP3. To decrease the cost of PhLA production and to avoid other factors influencing LAB fermentation, other reagents or buffer solutions for pH adjustment were not employed in this study (line 255-258). However, we agree with the Reviewer’s suggestion. In the further study, the initial pH will be adjusted to a suitable pH for the subsequent LAB fermentation.
- The relationship between the concentration of sugar Porphyra Residues and the LAB count should be shown more clearly.
The authors thank the Reviewer for pointing out the important issue. As shown in Figure 2, the amount of KP3 and KP4 increased and gradually decreased after fermentation of 8% Porphyra residues for 72 and 96 h, respectively. The number of bacteria obviously decreased after fermentation of 2% Porphyra residues for 12 h, revealing that 8% Porphyra residues provides much more nutrients and a fermentation environment with a suitable pH for LAB growth compared with 2% Porphyra residues. Moreover, the peak phase period of LAB cultured with 8% Porphyra residues was persistent from 6–120 h post-fermentation, which is much longer than that cultured with 2% Porphyra residues and that in MRS medium. Thus, 8% was determined as the highest ratio of Porphyra residues for LAB fermentation. (line 193-201). As shown in Figure 4, Porphyra residues pretreated with 100, 500, 1000, 2000, and 4000 U of cellulase produced approximately 0, 10, 20, 40, and 80 mg/mL of reducing sugars, respectively. Accordingly, the higher the level of cellulase employed, the greater the amount of total sugar and reducing sugar produced (line 238-242). As shown in Figure 6, approximately 8.2, 8.2, 8.5, 9.6, and 9.2 log CFU/mL of KP3 were detected after the fermentation of Porphyra residues pretreated with 50, 250, 500, 1000, and 2000 U of cellulase, respectively. The higher the level of cellulase employed for pretreatment, the higher the counts of LAB and the lower the pH value (Figure 6). Taken together with the results in Figures 2, 4, and 6, a positive correlation was observed between the concentration of Porphyra residues and the LAB counts (line 249-255).

Reviewer 2 Report
(1) Figures are too small and texts inside figures are hard to read.
(2) The reason(s) must be given why and how to choose reaction parameters (in particular pH and duration of reaction).
(3) Effects of reaction parameters need to be further elaborated.
(4) It would be good to add the process developed through this work compared to other similar processes reported in earlier literature.
(5) The word "biorefinery" may be deleted from the title.
(6) Elaborate what the process shown in this work can be distinguished from other similar processes.
Author Response
Response to the Reviewers' comments:
We have studied carefully the comments from the Reviewers and revised the manuscript by taking all issues mentioned in the Reviewers into consideration. We highlighted all the revisions in red in the revised manuscript. The revised manuscript has undergone English language editing by MDPI. The text has been checked for correct use of grammar and common technical terms, and edited to a level suitable for reporting research in a scholarly journal. Following is a detailed point-by-point explanation on how we have addressed all the Reviewers’ concerns. The Reviewer’s comments are in black, and our responses are in blue.
Reviewer 2:
- Figures are too small and texts inside figures are hard to read.
The authors thank the Reviewer for the valuable comments. All of the figures were re-organized for clear texts and figures shown in the revised manuscript.
- The reason(s) must be given why and how to choose reaction parameters (in particular pH and duration of reaction).
The authors thank the Reviewer for pointing out the important issue. The experiments were accomplished according to a recent report [12] with some modifications to comprehensively investigate the impact of reaction parameters on LAB fermentation and PhLA production. In this study, Porphyra residues were pretreated with 0, 50, 100, 500, 1000, 2000, or 4000 U cellulase AP3 at 37 °C for 0–48 h for enzymatic saccharification. For fermentation, pretreated or untreated Porphyra residues (2%, 4%, or 8%) were added with or without UF < 100 kDa (8.21%), Phe (1.36% or 6.8%), and/or yeast extract (0.5% or 2.5%), and inoculated with cultivated KP3 or KP4 (1%; v/v) for 228 h. The cultivated medium and cell-free supernatants were collected every 6 h to determine the LAB count and pH value. After centrifugation at 12,000 ×g for 20 min, the supernatants were lyophilized for extraction and determination of PhLA, lactic acid, and Phe. To clearly show the results in the limited space, not all of the data collected every 6 h during the period of fermentation are shown in the following figures. The above description is added in the revised manuscript (line 103-115).
- Effects of reaction parameters need to be further elaborated.
The authors thank the Reviewer for pointing out the important issue. Effects of reaction parameters, including the levels of Cellulase, Porphyra residues, Phe, yeast extract, fermentation time and pH value, are elaborated in the revised manuscript.
As shown in Figure 2, the amount of KP3 and KP4 increased and gradually decreased after fermentation of 8% Porphyra residues for 72 and 96 h, respectively. The number of bacteria obviously decreased after fermentation of 2% Porphyra residues for 12 h, revealing that 8% Porphyra residues provides much more nutrients and a fermentation environment with a suitable pH for LAB growth compared with 2% Porphyra residues. Moreover, the peak phase period of LAB cultured with 8% Porphyra residues was persistent from 6–120 h post-fermentation, which is much longer than that cultured with 2% Porphyra residues and that in MRS medium. Thus, 8% was determined as the highest ratio of Porphyra residues for LAB fermentation. (line 193-201).
As shown in Figure 3D, the concentration of lactic acid and the pH value did not significantly change during the fermentation period. Phe was not yet completely converted, suggesting that the ratio of added Phe and yeast extract provided sufficient PhLA precursor and nitrogen for KP3 fermentation. To further evaluate the impact of Phe and yeast extract on PhLA production, the concentrations of Phe and yeast extract were increased to 6.8% and 2.5%, respectively. However, the growth of LAB strains was not observed (data not shown), probably due to high osmotic pressure (line 217-224).
As shown in Figure 4, Porphyra residues pretreated with 100, 500, 1000, 2000, and 4000 U of cellulase produced approximately 0, 10, 20, 40, and 80 mg/mL of reducing sugars, respectively. Accordingly, the higher the level of cellulase employed, the greater the amount of total sugar and reducing sugar produced (line 239-242).
As shown in Figure 6, approximately 8.2, 8.2, 8.5, 9.6, and 9.2 log CFU/mL of KP3 were detected after the fermentation of Porphyra residues pretreated with 50, 250, 500, 1000, and 2000 U of cellulase, respectively. The higher the level of cellulase employed for pretreatment, the higher the counts of LAB and the lower the pH value (Figure 6). Taken together with the results in Figures 2, 4, and 6, a positive correlation was observed between the concentration of Porphyra residues and the LAB counts. Notably, the pH value of commercial cellulase AP3 is 4.5. Therefore, a lower pH in Porphyra residues was observed when pretreated with more cellulase AP3. To decrease the cost of PhLA production and to avoid other factors influencing LAB fermentation, other reagents or buffer solutions for pH adjustment were not employed in this study (line 249-258).
- It would be good to add the process developed through this work compared to other similar processes reported in earlier literature.
The authors thank the Reviewer for pointing out the important issue. So far, no study pertaining to the production of PhLA from algae has been published. With respect to the use of biomass, only one study used dilute acid-pretreated sorghum bagasse as a lignocellulosic feedstock for PhLA production using a recombinant Escherichia coli strain expressing phenylpyruvate reductase. However, the PhLA yield was reduced by 35% compared with the filter paper hydrolysate during fermentation with the enzymatic hydrolysate of sorghum bagasse as a carbon source [21]. Recently, we developed an enzyme-assisted method for cell-wall disruption and PE and PC extraction from Porphyra. However, residual Porphyra biomass and UF eluate (UF < 100 kDa) were yielded. In this study, we applied Porphyra residues and UF < 100 kDa to harvest PhLA, a valuable biopreservative, via LAB fermentation. By screening the strains of LAB and modifying the formula of the carbon and nitrogen source, PhLA precursor, cellulase, and fermentation time, the process for PhLA production was optimized. Up to 2.5 times higher the amount of PhLA was harvested than that produced from KP3 cultured in MRS broth. Taken together, integration of upstream phycobiliproteins extraction and downstream PhLA production from Porphyra provides a novel biorefinery process to enhance eco-friendly Porphyra-based bioeconomy (line 295-310). Therefore, we believe that this is a novel study for the consideration of publication.
- The word "biorefinery" may be deleted from the title.
The authors thank the Reviewer for this suggestion. The title of this study is revised as “Production of Phenyllactic Acid from Porphyra Residues by Lactic Acid Bacteria Fermentation” (line 2-3).
- Elaborate what the process shown in this work can be distinguished from other similar processes.
The authors thank the Reviewer for pointing out the important issue. So far, no study pertaining to the production of PhLA from algae has been published. With respect to the use of biomass, only one study used dilute acid-pretreated sorghum bagasse as a lignocellulosic feedstock for PhLA production using a recombinant Escherichia coli strain expressing phenylpyruvate reductase. However, the PhLA yield was reduced by 35% compared with the filter paper hydrolysate during fermentation with the enzymatic hydrolysate of sorghum bagasse as a carbon source [21]. Recently, we developed an enzyme-assisted method for cell-wall disruption and PE and PC extraction from Porphyra. However, residual Porphyra biomass and UF eluate (UF < 100 kDa) were yielded. In this study, we applied Porphyra residues and UF < 100 kDa to harvest PhLA, a valuable biopreservative, via LAB fermentation. By screening the strains of LAB and modifying the formula of the carbon and nitrogen source, PhLA precursor, cellulase, and fermentation time, the process for PhLA production was optimized. Up to 2.5 times higher the amount of PhLA was harvested than that produced from KP3 cultured in MRS broth. Taken together, integration of upstream phycobiliproteins extraction and downstream PhLA production from Porphyra provides a novel biorefinery process to enhance eco-friendly Porphyra-based bioeconomy (line 295-310).

Round 2
Reviewer 1 Report
The authors answered my questions acceptably and I think it is publishable in the current version.
Reviewer 2 Report
The authors addressed the reviewer's comments adequately.